# Entomological Surveillance in Former Malaria-endemic Areas of Southern Italy

**DOI:** 10.3390/pathogens10111521

**Published:** 2021-11-21

**Authors:** Donato Antonio Raele, Francesco Severini, Daniela Boccolini, Michela Menegon, Luciano Toma, Ilaria Vasco, Ettore Franco, Pasquale Miccolis, Francesco Desiante, Vincenzo Nola, Pietrangelo Salerno, Maria Assunta Cafiero, Marco Di Luca

**Affiliations:** 1Laboratorio di Entomologia Sanitaria, Istituto Zooprofilattico Sperimentale della Puglia e della Basilicata, 71121 Foggia, Italy; donatoantonio.raele@izspb.it (D.A.R.); ilaria.vasco@izspb.it (I.V.); mariaassunta.cafiero@izspb.it (M.A.C.); 2Dipartimento Malattie Infettive, Reparto Malattie Trasmesse da Vettori, Istituto Superiore di Sanità, 00161 Rome, Italy; francesco.severini@iss.it (F.S.); daniela.boccolini@iss.it (D.B.); michela.menegon@iss.it (M.M.); luciano.toma@iss.it (L.T.); 3Dipartimento di Prevenzione, Azienda Sanitaria Locale, 74121 Taranto, Italy; ettore.franco@asl.taranto.it (E.F.); pasquale.miccolis@asl.taranto.it (P.M.); francesco.desiante@asl.taranto.it (F.D.); 4Dipartimento di Prevenzione, Sanità e Benessere Animale, Azienda Sanitaria Locale, 75100 Matera, Italy; vnola@asmbasilicata.it (V.N.); pietrangelo.salerno@asmbasilicata.it (P.S.)

**Keywords:** malaria, *Anopheles*, residual anophelism, Apulia, Basilicata

## Abstract

Malaria still represents a potential public health issue in Italy, and the presence of former *Anopheles* vectors and cases imported annually merit continuous surveillance. In areas no longer endemic, the concurrent presence of gametocyte carriers and competent vectors makes re-emergence of local transmission possible, as recently reported in Greece. In October 2017, due to the occurrence of four suspected introduced malaria cases in the province of Taranto (Apulia region), entomological investigations were performed to verify the involvement of local anopheline species. In 2019–2020 entomological surveys were extended to other areas historically prone to malaria between the provinces of Taranto and Matera and the province of Foggia (Gargano Promontory). Resting mosquitoes were collected in animal shelters and human dwellings, larvae were sampled in natural and artificial breeding sites, and specimens were both morphologically and molecularly identified. A total of 2228 mosquitoes were collected, 54.3% of which were anophelines. In all the investigated areas, *Anopheles labranchiae* was the most widespread species, while *Anopheles algeriensis* was predominant at the Gargano sites, and *Anopheles superpictus* and *Anopheles plumbeus* were recorded in the province of Matera. Our findings showed a potentially high receptivity in the surveyed areas, where the abundance of the two former malaria vectors, *An. labranchiae* and *An. superpictus*, is related to environmental and climatic parameters and to anthropic activities.

## 1. Introduction

In recent years, the increase in globalization coupled with climatic and human-induced environmental changes have raised concerns about the possible introduction or reintroduction of vector-borne diseases, including malaria, into Europe [1,2,3].

In countries no longer endemic, primarily those bordering the Mediterranean basin where malaria was eradicated, the concurrent presence of gametocyte carriers and competent mosquitoes may not only favor the re-emergence of sporadic autochthonous cases but may also make local transmissions sustainable, as recently reported in Greece [4,5].

In the past, malaria transmission in Italy was associated primarily with the presence of two vector species, *Anopheles labranchiae* Falleroni, 1926, and *Anopheles sacharovi* Favre, 1903, both belonging to the *Anopheles maculipennis* Meigen complex. The first species, which is still present in Italy in scattered foci, was considered the main vector along the central and southern coasts and in Sicily and Sardinia. The second showed a more limited distribution, mainly along the upper and lower Adriatic coasts and in Sardinia [6,7]. Following the malaria eradication campaign (1947–1951), the abundance of *An. sacharovi* was greatly reduced, and this species has not been found in the country since the 1960s, probably because of the progressive disappearance and/or modification of its larval habitats [8,9,10,11].

Nevertheless, other species of the *An. maculipennis* complex, such as *Anopheles atroparvus* Val Thiel, 1927, have contributed to the continued existence of low levels of endemicity in some inland areas of the country, where the two main vectors were absent. In addition, *Anopheles superpictus*, which is not included in the *An. maculipennis* complex and is currently present in central and southern Italy and in Sicily has played a vector role in these areas in the past [12].

The reclamation of marshy areas and the concomitant use of DDT had already led to the interruption of *Plasmodium* transmission during the initial years of the malaria eradication campaign [13]. However, autochthonous cases of *P. vivax* malaria were reported in Sicily until 1962 [14]. In 1970, the World Health Organization (WHO) officially declared the country free from malaria [14].

Since then, most cases of malaria recorded in Italy and more generally in Europe have been imported, with a steadily growing number of cases driven by the intensification of international travel and migratory flows from endemic areas, and malaria is now ranked first among imported parasitic diseases [4,15].

The risk of reactivation of local transmission of malaria is linked to a combination of three concomitant factors: receptivity, which depends on the presence, abundance and biological behavior of anopheline vectors; vulnerability, i.e., the presence of imported reservoirs of infection (gametocyte carriers); and infectivity of vectors, which is in turn influenced by the genetic traits of vector species and by climatic, ecological and other favoring factors [16,17].

Some areas of the country, particularly in rural contexts of the central and southern regions, are more susceptible to this risk because they are characterized by both climatic and ecological conditions favorable to the development of malaria vectors [12]. Moreover, the presence of immigrants, moving from malaria-endemic countries and employed most often as farm workers, could create a risk for both immigrant and resident communities where infected individuals are present, particularly in southern regions. 

The National Surveillance System for Malaria, drawn up by the Ministry of Health and the National Institute of Health, aims to prevent the risk of indigenous transmission and, where necessary, adopt targeted, effective control measures [18]. One of the main requirements is, therefore, to have access to up-to-date monitoring of receptivity within the country, namely the presence and density of local *Anopheles* vectors, as well as their behavior and ecology.

Entomological surveys have been carried out since eradication in only one former hyperendemic area, the Maremma Plain (in the province of Grosseto in Tuscany), and these activities have provided useful information that has helped reconstruct the history of malaria and its vectors over past decades [19,20,21,22]. Moreover, several studies in previous years have provided an update on *An. maculipennis* complex distribution in other areas [7,9,22,23]. Investigations were also carried out recently in regions in northern Italy [24,25]. Conversely, little data are available for southern Italy, and the islands and the information that exists is often very dated [7,8,12,23,26,27,28,29,30,31]. 

Historically, the Apulia and Basilicata regions were endemic for malaria in both coastal areas and inland. The vectors involved were the primary efficient species, *An. labranchiae* and *An. sacharovi*. Particularly on the Gargano Promontory, these two species were sympatric, colonizing a wide range of brackish water collections widely present in these areas [11,31,32].

Recently, following four suspected locally transmitted malaria cases occurring in the region of Apulia in 2017, an epidemiological investigation was carried out [33,34]. Specifically, in late September and early October, four *Plasmodium falciparum* cases among immigrant agricultural laborers were reported in the province of Taranto, with all cases exhibiting the onset of symptoms in the same week and having declared no recent travel history to malaria-endemic countries. Despite the extensive epidemiological investigation, neither the potential source of the infection nor the mode of transmission was identified at the time [33,34]. However, the concomitant entomological survey identified both *An. labranchiae* in the area involved and *An. superpictus* in nearby sites in the province of Matera in the Basilicata region. This event clearly highlighted an epidemiological situation involving the potential risk of re-emergence of malaria transmission in these areas [33]. Further investigations were therefore planned to rule out the possibility that the vector densities found in the area could be compatible with the possibility of local transmission events generated by imported cases. In order to fill this important gap with regard to the scarce information on residual anophelism, an entomological investigation was undertaken, not only in the area affected by the suspected autochthonous malaria cases (the provinces of Taranto and Matera) in 2017 but also in those areas historically endemic for the disease, such as the Gargano Promontory (in the province of Foggia). The periods required for technical project implementation prevented any entomological activities in 2018, and it was only in the following two years (2019–2020) that data collection was possible.

The results of this entomological study carried out through these periodic surveys in 2017 and 2019–2020 are in this paper.

## 2. Results

During the 2017 and 2019–2020 surveys, a total of 2228 adult mosquitoes were collected at the selected sites. Of the specimens collected, 1209 (54.26%) were anophelines, of which 645 (53.35%) belonged to the *An. maculipenni*s complex.

Nineteen species belonging to six genera were identified in the total sample: *Aedes albopictus* (Skuse, 1894), *Aedes caspius* (Pallas, 1771), *Aedes detritus* (Haliday, 1833), *Aedes geniculatus* (Olivier, 1791), *Aedes vexan*s (Meigen, 1830), *Anopheles algeriensis* Theobald, 1903, *Anopheles plumbeus* Stephens, 1828, *Anopheles labranchiae* Falleroni, 1926, *Anopheles superpictus* Grassi, 1899, *Culex hortensis* Ficalbi, 1889, *Culex laticinctus* Edwards, 1913, *Culex pipiens* Linnaeus, 1758, *Culex territans* Walker, 1856, *Culex theileri* Theobald, 1903, *Culex univittatus* Theobald, 1901, *Culiseta annulat*a (Schrank, 1776), *Culiseta longiareolata* (Macquart, 1838), *Coquillettidia richiardii* (Ficalbi, 1899) and *Uranotaenia unguiculata* (Edwards, 1913).

Species composition and corresponding frequencies (%) by municipality and year of the collection are shown in Table 1.

Potential larval breeding sites around the selected sites were inspected during all entomological surveys, and the results are shown in Table 2.

As our study focused on *Anopheles*, the results of collections by site, year and capture method refer to this genus only (Table 3).

### 2.1. Molecular Analyses

More than 62% (n = 401) of the *An. maculipennis* sensu lato (sl) sample was analyzed using internal transcribed spacer 2 (ITS-2) sequencing, and the entire subsample was identified as *An. labranchiae* (ITS-2 sequence GenBank accession number OK021590-600). Within this subsample, 30 *An. labranchiae* females, previously identified using egg morphology (Figure 1), were included, and no diagnostic discrepancy was observed. 

No intraspecific variation was detected, and ITS-2 sequences shared 100% identity with the homologous sequences from the Apulia (AY253841), Tuscany (AY232827) and Sardinia (AY253840) regions, available in GenBank.

In addition, the mitochondrial cytochrome oxidase subunit 1 (COI) marker was also characterized for 10 *An. labranchiae* specimens, previously identified molecularly using ITS-2. Alignment of the obtained COI fragment (722 bp) showed a high level of intraspecific genetic diversity (up to 18 nucleotide-variable silent sites). The 10 sequences were separated by genetic distances ranging from 0.0028 to 0.0255. Four haplotypes, H1–4 (COI sequence GenBank accession numbers OK047734-7), were identified with nucleotide variability ranging between 95.5% and 99.7%. The BLAST analysis on a portion (464 nt) of *An. labranchiae* COI sequences yielded ambiguous results: haplotypes H1 (shared by two specimens) and H2 (by three) showed higher nucleotide identity (99.35% and 99.78%, respectively) with the HQ860355 sequence of *An. labranchiae*, whereas haplotypes H3 (shared by three specimens) and H4 (by two) showed higher identity (99.57% and 99.78%, respectively) with the MK402872 sequence of *An. atroparvus* available in GenBank.

Furthermore, to corroborate the morphological identification of *An. algeriensis*, the ITS-2 region of 63 specimens from sites 1 and 6 was also analyzed (ITS-2 sequence GenBank accession numbers OK030903-4). The *Anopheles algeriensis* sample showed no intraspecific variability and about 96% identity with 13 sequences from Spain available in GenBank (MK412727-32, MK412734, MK412745, MK412751, MK412753, MK412757-58, MK412760).

### 2.2. Anopheline Species Diversity and Distribution

#### 2.2.1. Entomological Survey after the Four Suspected Indigenous Cases (2017)

In October 2017, immediately after the alert on the four cases of suspected introduced malaria in the municipality of Ginosa, in loc. Girifalco (TA), entomological investigations were quickly carried out, starting with the dwelling where the African migrants were living (site 14). There was no livestock, and only a few dogs were present. On that occasion, a female of *An. maculipennis* sl was collected using a CDC-light trap (BioQuip Products, Rancho Dominguez, CA, U.S.A.), and later, a male of *An. maculipennis* sl was found in a dog kennel close to the house. Both specimens were identified as *An. labranchiae* by molecular analyses.

The investigation continued with an inspection of site 13, which was very close to site 14, and two other farms (sites 8 and 9) (Figure 2).

A total of 103 mosquitoes were captured, and all were identified molecularly as *An. labranchiae*. In addition, at site 8, an inspection of potential larval breeding sites revealed some *Anopheles* larvae (with a larval density ranging from 0.01 to 0.1 larvae per dip) along a slow-flowing ditch rich in vegetation. The larval specimens were later identified as *An. labranchiae.*

That entomological survey also included site 17, a farm near site 8 but in the adjacent province of Matera (Basilicata region). Here, the CDC trap collected two *An. superpictus* females.

In order to determine the distribution and density of these *Anopheles* species, visits were also made to two other farms (sites 15 and 16) in the Matera province, located a short distance (3–8 km) from the Basento river. In summer, water scarcity generates pools along the stony riverbed, which potentially serve as larval foci for the mosquito. At site 16, three *An. superpictus* females were collected, and at site 15, 64 *An. superpictus* specimens and 36 *An. maculipennis* sl specimens were captured. A sample (36%) of the latter species was molecularly analyzed and identified as *An. labranchiae* (Table 3).

#### 2.2.2. Entomological Surveillance Activities (2019–2020)

In 2019, the study was limited to the Ginosa municipality (TA), and in October, sites 8, 9 and 13 were visited again, and the situation identified two years earlier was confirmed: of 60 specimens of *An. maculipennis* sl, 49 (81.6%) were identified as *An. labranchiae*. Moreover, the study area was expanded to include new sites (10, 11 and 12): at the first two sites, two females of *An. labranchiae* were collected (at site 11, a mosquito trying to land on one of the collectors), while site 12 was negative (Table 3).

Entomological activities in 2020 initially focused on two farms (sites 15 and 16) that had already been investigated in 2017. Site 15 was inspected in July and 26 *An. maculipennis* sl were collected using both aspirators and traps. Of those specimens, 85% were identified as *An. labranchiae*. Site 16 was visited in both June (30 *An. labranchiae* collected) and July, when 270 specimens of *An. maculipennis* sl were captured, showing a statistically significant difference between two collection periods (*p* < 0.05) due to more favorable weather variables. About 23% of the sample was molecularly analyzed and found to be *An. labranchiae*. On this occasion, larvae were collected by dipping in ponds and pools all around the site (0.1–1.9 larvae/dip), and the larvae were *An. labranchiae*. In addition, *An. labranchiae* larvae were also captured and identified at different points along the banks of the nearby Basento river, where the larval density ranged from 0.05 to 1.3 larvae/dip. No specimens of *An. superpictus* were collected. Along the road between the two farms, the Salandra wood (site 18) was also visited, and one *Anopheles plumbeus* was caught while it was attempting to bite humans. Meanwhile, several larvae of the species were collected inside a hollow oak tree, which was filled with water rich in plant matter.

Furthermore, in July and September 2020, an entomological investigation was carried out in a new area in Apulia, the Gargano promontory. During the survey, *Anopheles algeriensis* was detected mainly at site 1 (over 96% of mosquitoes collected), near Lake Lesina, at a buffalo farm not subject to intensive farming practices. Here, the mosquito species were captured directly by aspirators and CDC traps, both indoors and outdoors. Specimens of *An. maculipennis* sl were also collected and molecularly identified as *An. labranchiae* (>97% of the sample). Both sites 2 and 3 were positive for the presence of *An. labranchiae*, whereas site 4 was negative. At site 6, near Lake Salso, a CDC trap placed outside a donkey shelter collected *An. algeriensis* females. At the same site, specimens of *An. maculipennis* sl were also collected and then molecularly identified as *An. labranchiae* (about 94% of the sample from the site). Furthermore, the number of *An. maculipennis* sl collected at this site was significantly more relevant than at site 1, which is characterized by a greater anthropogenic impact than site 6 (*p* < 0.05). At site 5, seven of the nine *An. maculipennis* sl females collected were identified as *An. labranchiae*. At site 7, in the Umbra Forest (municipality of Monte Sant’Angelo) located in the hinterland of the Gargano promontory, no *Anopheles* were found (Table 3).

In the Gargano area, the search for *Anopheles* larvae was negative. All potential larval breeding sites, irrigation and drainage canals and marshy banks and lagoons (sites 1, 2, 5 and 6) were inspected without success. 

However, a significant percentage (76.2%, *p* = 0.0001) of *An. maculipennis* sl males collected at resting site 6 was highly suggestive of the presence of suitable larval habitats in the surrounding area, considering the limited dispersal of mosquito males. Of note, the number of *An. algeriensis* males collected at site 1 (about 5% of the sample), taking into account the exophilic behavior of the species.

## 3. Discussion

This study on residual anophelism, initially planned in the area where the four suspected introduced malaria cases (October 2017) occurred, was extended to other former endemic-malaria areas in the two-year period 2019–2020, thanks to the funding of the Ministry of Health. By taking into account the phenology of mosquito species potentially present, the research project involved a series of investigations with the aim of updating the fauna of mosquitoes, particularly of *Anopheles* occurring in the study areas, with the limits associated with this type of occasional collection.

According to our findings, foci of anophelines were found all over the study area, in both of the regions of Apulia and Basilicata, during the entomological investigations carried out in 2017 and 2019–2020. However, species composition and population density showed differences related to the variety of biotopes and human activities that characterized the selected sites. Numerous larval breeding places around the adult collection sites were identified during all of the entomological investigations, but only a few of these were positive for the presence of *Anopheles* larvae.

With regard to molecular identification of *Anopheles* species, our results confirmed the discriminatory power of the ITS-2 marker for species complexes, with all ITS-2 sequences obtained from *An. maculipennis* sl specimens being referred to a single haplotype of *An. labranchiae*. Conversely, the COI fragment produced a significant degree of intrapopulation polymorphism, generating four haplotypes from 10 *An. labranchiae* specimens that were collected in the same area. In interpreting these results, we should consider the insufficient resolution of this molecular marker, which is therefore ineffective in discriminating among members of the *An. maculipennis* complex [35]. Moreover, for *An. algeriensis* the morphological identity was confirmed using the ITS-2 as a molecular target.

*Anopheles algeriensis* was the second prevalent species in numerical terms, not only among all mosquito species (more than 22%) but also among the *Anopheles* species collected (about 41%). In Italy, this thermophilic mosquito, once very common along southern coastal areas and in Sicily and Sardinia [36], has become much rarer due to the progressive reduction in habitats for the development of larvae [37,38]. However, up until the 1970s, *An. algeriensis* was reported in the Peschici area, along the Gargano coast [27]. The species grows in freshwater but can also tolerate a certain degree of salinity. Although the larvae can also be collected in wells and cisterns, typical reproduction sites are banks along swamps and marshes, ponds, pools, low areas of lakes and bends of low-flowing streams, generally shaded by rich vegetation (*Juncus* and *Phragmites* spp.) [39]. In Italy, these environments were shared in the past with other *Anopheles* mosquitoes, such as *An. labranchiae* and *An. sacharovi* [36]. Aitken [40] evaluated the seasonal dynamics of *An. algeriensis* in Sardinia, showing a larval peak in late summer and early autumn. As it is a mosquito that is both anthropophilic and zoophilic [41], this species is thought to have a certain competence in the transmission of human Plasmodia [39], even though it has never played a significant role in Italy. However, the possible involvement of this species in the transmission of *Plasmodium malariae* was hypothesized in Sardinia [42].

Found in July and September 2020, *An. algeriensis* showed in this study a localized distribution along the coast, east and west of the Gargano promontory, near large wetlands such as Lake Lesina and Lake Salso (sites 1 and 6, respectively). Unfortunately, despite efforts to collect immature stages of such species, it was not possible to detect its larval foci. At site 1, this could be because the larval density in the nearby irrigation canal was too low, although the presence of males (about 5% of the sample) suggests a breeding site very close by. Lake Salso (site 6) is one of the largest and most important coastal wetlands in southern Italy and is part of the Gargano National Park. Once entirely marshy, this area has been reclaimed and is now managed by the WWF. It consists of pastures, flooded meadows, marshes and a large lagoon, where birdwatching towers, buildings, a restaurant and animal shelters are located. Here, the large size of the area to be monitored (over 1000 ha) and, above all, the presence of larvivorous fish (*Gambusia* sp.) undoubtedly prevented the detection of the larvae of the species. *Anopheles algeriensis* is most active at dusk and dawn, and its behavior is primarily exophagic and exophilic [39], as confirmed by our collection operations using outdoor CDC traps and by the fact that mosquito adults were never found at rest in animal shelters. 

*Anopheles labranchiae* was historically the primary malaria vector throughout the country. Over time, this species has been significantly affected by anthropogenic environmental changes that have reduced its footprint. Currently, this mosquito species is present along the central and southern coastal strips and on the major islands of Sicily and Sardinia, in scattered and discontinuous foci [10,31,37]. *Anopheles labranchiae* can still achieve epidemiologically relevant densities where ecological conditions allow the mosquito to reproduce substantially during the warm season, as occurs in the rice fields of the Maremma Plain in Tuscany [20,21].

In Apulia, *An. labranchiae* and *An. sacharovi*, the former malaria vectors, were still present along the Gargano coast even after the anti-malaria campaign that followed the Second World War. In particular, the two mosquito vectors were found in sympatry in various localities of the Gargano promontory near Vieste, Peschici, Lesina and Cagnano Varano [31,36]. Subsequent reclamation activities (through the diking and draining of marshes and retrodunal ponds, but above all, through the large-scale use of insecticides, once sprayed by airplanes), urbanization and pollution have reduced many of the larval habitats, especially for *An. sacharovi*, though this species was still reported in the 1970s in these locations [27]. During the summer of 1993, Romi et al. [31] carried out an entomological study along the entire coastal strip of the Gargano promontory, recording the presence only of *An. labranchiae* in the area of Lesina, Varano and Vieste and in the Candelaro area north of Lake Salso. This mosquito species was found in sporadic foci, in some cases with non-significant densities (0.01–0.1 larvae/dip and 20–30 adults per animal shelter). *Anopheles sacharovi* has not been found since that time, as many of its natural and typical breeding sites have disappeared. The latest entomological research, conducted 10 years later, also confirmed the disappearance of *An. sacharovi* and a clear reduction in the distribution of *An. labranchiae*, both in Apulia and in Basilicata. Although 52 sites were investigated in the provinces of Foggia, Brindisi, Bari, Lecce and Taranto and seven sites in the province of Matera, *An. labranchiae* was found only in the Lesina area [23]. Human and environmental factors, such as land use and the presence of water and vegetation, along with climatic variables, such as temperature and rainfall, may have had an impact on the disappearance of *An. sacharovi*, being more favorable for more thermophilic and better-adapted species, i.e., *An. labranchiae*.

With respect to this study, our results show *An. maculipennis* sl (*An. labranchiae* included) as the predominant (about 29%) and most widespread taxon, considering all mosquito species identified in the study area. In addition, this taxon represented more than 53% of all *Anopheles* collected during the study. The data analysis also suggests that all specimens belonging to *An. maculipennis* complex can presumably be ascribed to *An. labranchiae*, as evinced by the molecular diagnosis of over 62% of the entire sample analyzed. This study confirms the presence of *An. labranchiae* on the Gargano promontory. Specifically, near Lesina and Cagnano Varano, *An. labranchiae* populations were not characterized by any relevant densities, while higher densities were detected in the Manfredonia area, near Lake Salso (*p* < 0.05). 

In the Ionian area of Apulia (municipality of Ginosa) and both along the coast and in the most inland areas of Basilicata, *An. labranchiae* populations were collected in all the municipalities investigated (except Bernalda), in all cases at sporadic foci but in epidemiologically significant densities.

*Anopheles superpictus*, once a vector of malaria in central and southern Italy and Sicily, currently shows a discontinuous distribution that has progressively diminished over time. Decreases in the density of this species were recorded by Romi et al. [31] in Calabria, along both the Ionian and Tyrrhenian coasts, compared to 15 years earlier [28,29]. A more recent finding of a few *An. superpictus* specimens date back to 2011 in Basilicata [43]. The pollution of lakes and rivers and the use of water for agricultural and industrial purposes have, in fact, considerably reduced the larval habitats of the species. Larval breeding sites consist of shallow pools of water that form in the stony beds of rivers and streams during periods of summer drought. This *Anopheles* exhibits a summer–autumn phenology that reaches its maximum density between August and September. A marked endophily, combined with a high degree of anthropophily, allows the mosquito to feed not only on cattle but also on humans when available.

In this study, *An. superpictus* was recorded only in 2017 in Basilicata, representing over 3% of the Culicidae fauna and 5.7% of anophelines. In particular, numerous specimens of *An. superpictus* were found at the most inland site in the area surveyed (the municipality of Grottole), in sympatry with *An. labranchiae*. The species exhibited very low densities at the other two collection sites (the municipalities of Ferrandina and Bernalda), but *An. labranchiae* was only also identified at Ferrandina. No larvae of *An. superpictus* were found. With regard to its absence in 2020, this could be traced back to its phenology and, therefore, a difference in seasonal density: in 2017, entomological collection activities were carried out in the second half of October, while three years later, in 2020, they took place in the second week of July.

*Anopheles plumbeus* thrives mainly in rainwater collections within holes in tall trees, such as *Platanus*, *Ulmus* and *Quercus*, often in association with other dendrolimnic species (e.g., *Aedes geniculatus, Aedes berlandi* and *Orthopodomyia pulcripalpis*), not only in wooded areas but also along tree-lined avenues and in gardens in cities. Unlike other *Anopheles*, this species lays its eggs on dry substrates above the surface of the water, and they only hatch when submerged. Its ecological plasticity means that this mosquito colonizes even artificial containers filled with water, a factor that encourages huge densities of urban populations. *Anopheles plumbeus* is widely distributed throughout Italy, where there are suitable habitats for larval development. This species is characterized by diurnal trophic activity, is exophagic and exhibits very aggressive behavior, feeding mainly on humans and other mammals. Both in the past and in recent times, the species has been suspected of being an occasional vector of malaria in large urban centers, especially in Northern Europe [44,45]. In 2020, *An. plumbeus* was found in Basilicata in a wooded area (in Salandra Wood in the province of Matera), both as adult and larvae, in its typical natural environment. Conversely, the species was not detected in the Umbra Forest in Apulia, where it was previously reported [27].

In summary, this study, which began with the detection of four suspected introduced cases of *Plasmodium falciparum* in 2017, provides a detailed update on the mosquito fauna and, in particular on the *Anopheles* species in some areas of southern Italy, filling a gap in the data gathered in the last fifteen years. Moreover, these findings provide new insights into the distribution and ecology of potential malaria vectors, particularly those of the species belonging to the *Anopheles maculipennis* complex, which occur widely all over Europe. Furthermore, the use of the ITS-2 marker enabled the identification of *An. labranchiae* among the members of the complex, but also corroborated the morphological identification of *An. algeriensis*, a species that is considered very common in southern Italy but has rarely been found in recent entomological collections.

In the context of epidemiological surveillance in European non-endemic areas with high receptivity and low vulnerability, these results may provide useful information in terms of planning and implementing targeted malaria surveillance plans. The primary approaches and activities for preventing malaria reintroduction in temperate areas are represented by the management of imported cases and entomological surveillance, such as monitoring numbers of adult *Anopheles* mosquitoes and, potentially, vector control activities.

## 4. Materials and Methods

### 4.1. Study Area

The study areas fall within the regions of Apulia and Basilicata (southern Italy). In these areas, eighteen sites belonging to 10 municipalities in three provinces were selected for entomological surveillance (Figure 2); they were chosen on the basis of their ecological features and human activities. Most of the sites are farms with livestock and animal shelters since the presence of animals represents a strong attraction for mosquitoes such as *Anopheles,* and it is easier to collect them where they feed and rest.

In Apulia, two distinct geographical contexts were chosen, as they were once affected by endemic malaria: the Gargano promontory, to the north (Figure 2A), and the Ionian coast, on the border with Basilicata, to the south (Figure 2B). 

The Gargano area is in the province of Foggia (FG), overlooking the Adriatic Sea. A karst massif (about 1000 m a.s.l.) characterizes this area, and there are no surface water basins in the hinterland. The inland section of the promontory is partly covered by a beech forest, woodland and pasture, whereas lagoons (Lake Varano) and lakes containing brackish (Lake Lesina) or fresh water (Lake Salso), and banks and shores characterize the coastal sites, covered by shrubs and rich riparian vegetation. The area features a typical Mediterranean climate, with a yearly average temperature of 15.5 °C, ranging from a minimum of 3 °C in January to a maximum of 28 °C in July, and a yearly average rainfall of 58.75 mm (from 29 mm in August to 83 mm in November). Here, seven sites (1–7) were selected: six were farms near one of these water bodies, where historically *An. labranchiae* and *An. sacharovi* were widespread. Site 7 was chosen in a natural forest, where we could expect to find mosquito species typical of woodland environments, less common than urban or rural areas (Figure 2A).

The southern part of the region, fronting the Ionian Sea, belongs to the province of Taranto (TA). This territory ranges from the plains at sea level to a landscape of high hills, and it is intersected by a series of ravines of fluvial–karst origin. Vineyards alternate with olive trees, and there is an extensive pine forest of Mediterranean scrub shrubs running alongside the sandy beach. Climatic conditions are similar to the nearby province of Matera in Basilicata. Seven sites (8–14) were selected; site 14 was chosen because it was the temporary home of three of the four cases of suspected introduced malaria and thus the base site for the first entomological survey commenced in October of 2017; the other six sites were farms (Figure 2B).

The Basilicata area has a wide variety of natural biotopes, including Mediterranean bush, woodland, pasture and characteristic clay dunes known as ‘Calanchi’. Rivers, lagoons, ponds and lakes make up the regional hydrological system. The area is characterized as having a Mediterranean climate along the coast and a continental climate with winter snowfalls on inland mountains. The yearly average temperature is 26 °C, ranging from a minimum of −1 °C in January to a maximum of 29 °C in August, and yearly average rainfall is 141.5 mm (from 30 mm in May to 253 mm in December). In the administrative province of Matera (MT), four sites (15–18) were selected. Site 17 was a farm very close to the coastal sites of the adjacent province of Taranto. The inland sites 15 and 18 were farms located at altitudes of around 450 m and close to the main stretch of the Basento river. This river, similar to others in the area, features an exclusively rain-driven regime with large-scale floods in autumn and winter and extremely low water levels in summer, which create pools of water, typical larval foci for *An. superpictus*. Site 16 was in a wooded environment (Figure 2B).

The sites selected and their geographical coordinates and farming activities, and the period of mosquito collection are shown in Table 4.

### 4.2. Mosquito Collections

Entomological surveys (adult and larval collection) were carried out in 2017 and 2019–2020 (Table 4) to investigate the presence and abundance of mosquito species. Collection activities were performed using different and concurrent methods, as described in Di Luca et al. [20]. Battery-powered aspirators were used to actively collect resting mosquito females in animal shelters (cowsheds, horse stables, pigsties, sheep/goat pens, henhouses) and in human facilities (milking rooms, haylofts, garages and verandas). CDC-light traps (BioQuip Products, Rancho Dominguez, CA, U.S.A.) were activated in selected places from sunset to 9:00 a.m. On two fortuitous occasions, human bait collections were carried out in selected locations.

Potential breeding sites were inspected, with consideration given to those located not more than 500 m from the farms (irrigation canals, drainage canals, ditches, ponds and banks of rivers and lakes), along with a whole series of artificial containers within the farms that could contain water, even in small quantities. Larvae were sampled using a standard 350 mL dipper both at natural and artificial breeding sites. Streams, ponds and irrigation and drainage canals were sampled using 10–20 dips, depending on their size.

### 4.3. Laboratory Processing

Adult and larval specimens were identified according to morphological keys [37,38] and stored at -20 °C. A fraction of gravid females belonging to the *An. maculipennis* complex was induced to lay eggs to identify species by observation of exochorion ornamentation [46].

A representative number of adults of *An. maculipennis* sl and *An. algeriensis* were analyzed molecularly by using PCR and ITS-2 sequencing of ribosomal DNA [47]. Specifically, in 2017, all the specimens of *An. maculipennis* sl by site collected during the entomological survey that followed the reporting of the four cases of suspected introduced malaria were molecularly analyzed. In the two-year period 2019–2020, all collection sites were taken into consideration through molecular analysis of a number of *Anopheles* mosquitoes (from 100% to not less than 30%), depending on the number of specimens in the sample.

PCR reactions were performed using two mosquito legs placed directly in the PCR mix. Alternatively and exclusively for negative samples, the amplification reaction was performed using 5 μl of DNA extracted from the rest of the mosquito’s body using the PureLink Genomic DNA Kit or the Genejet Genomic DNA Purification Kit (ThermoFischer Scientific). All PCR products were purified and directly sequenced at Eurofins Genomics (Ebersberg, Germany) using the same primers as those used for PCR in both forward and reverse directions. The obtained sequences were compiled using DS Gene v1.5 software (Accelrys Inc. 2003) and analyzed using NCBI’s Basic Local Alignment Search (BLAST) for the identification of mosquito species.

For a subset of *An. labranchiae* specimens, a portion of COI gene was also characterized using primers described in Folmer et al. [48]. Genetic distance for the COI gene was calculated using the Kimura two-parameter (K2P) model [49].

### 4.4. Statistical Analysis

The differences between the sex proportions of *An. labranchiae,* as well as the abundance of anopheline species in relation to the ecological conditions of the collection sites, were evaluated by the chi-square test. Where appropriate, a comparison of the differences between the average number of mosquitoes collected per day was performed using the unpaired t-test. P values < 0.05 were considered statistically significant. All statistical analyzes were performed using a standard software package (Stata, version 14.0; StataCorp).

## 5. Conclusions

Our findings show the significant receptivity of the investigated areas, where former malaria vectors *An. labranchiae* and *An. superpictus* can be found at different densities depending on the kind of environment, climatic parameters and anthropic activities. In addition, the entomological surveys seem to confirm the disappearance of *An. sacharovi* in the areas surveyed. However, it is not possible to exclude the existence of residual foci of this mosquito species in modest densities, which could remain in particularly protected or isolated areas and which should be investigated in the near future.

## Figures and Tables

**Figure 1 pathogens-10-01521-f001:**
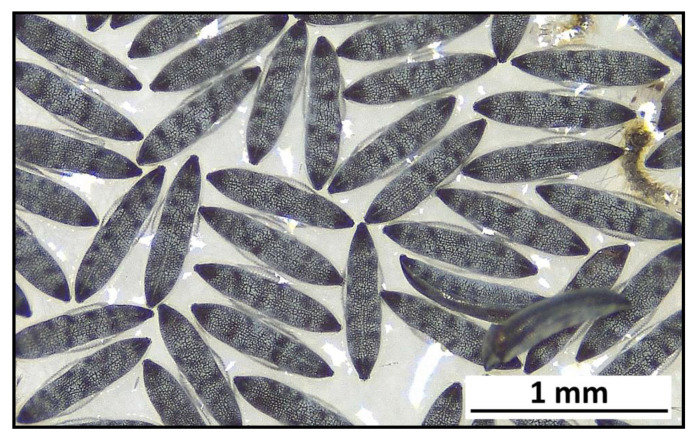
Typical eggs of *Anopheles labranchiae* with surface richly patterned, with wedge-shaped dark marks on a pale background and short and narrow floats.

**Figure 2 pathogens-10-01521-f002:**
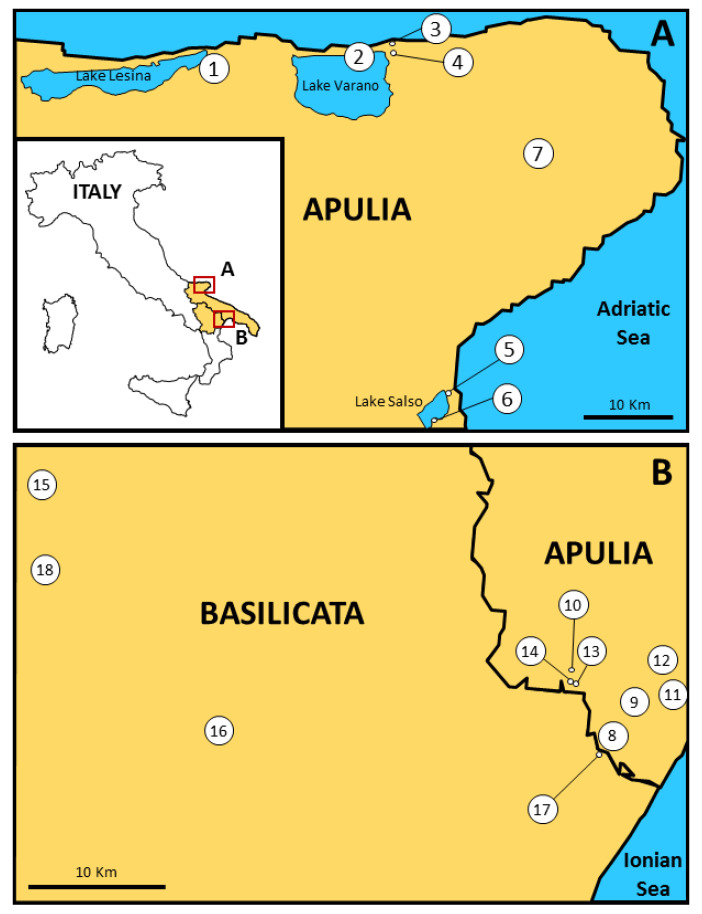
Location of study areas, indicating eighteen mosquito collection sites. (**A**) Sites on the Gargano promontory in the province of Foggia (Apulia); (**B**) Sites in the provinces of Taranto (Apulia) and Matera (Basilicata).

**Table 1 pathogens-10-01521-t001:** Number of adult mosquito species and corresponding frequencies (%) by municipality and month and year of collection.

	APULIA							BASILICATA							
	Lesina (FG)	Cagnano Varano (FG)	Ischitella (FG)	Manfredonia (FG)	Monte Sant’Angelo (FG)	Ginosa (TA)		Grottole (MT)		Ferrandina (MT)		Bernalda (MT)	Salandra (MT)		
**Species**	2020 (Jul, Sep)	2020 (Sep)	2020 (Sep)	2020 (Sep)	2020 (Jul)	2017 (Oct)	2019 (Oct)	2017 (Oct)	2020 (Jul)	2017 (Oct)	2020 (Jun, Jul)	2017 (Oct)	2020 (Jul)	tot	%
*Anopheles algeriensis*	475	-	-	19	-	-	-	-	-	-	-	-	-	494	22.17
*Anopheles labranchiae*	37	1	-	66	-	103	51	26	22	4	91	-	-	401	18.00
*Anopheles maculipennis* sl	1	1	-	6	-	-	11	10	4	2	209	-	-	244	10.95
*Anopheles plumbeus*	-	-	-	-	-	-	-	-	-	-	-	-	1	1	0.04
*Anopheles superpictus*	-	-	-	-	-	-	-	64	-	3	-	2	-	69	3.10
*Aedes albopictus*	1	2	-	-	1	-	4	-	-	-	1	-	-	9	0.40
*Aedes caspius*	15	12	178	40	-	1	4	-	-	-	-	-	-	250	11.22
*Aedes geniculatus*	-	-	-	-	-	-	-	-	-	-	-	-	1	1	0.04
*Aedes* spp.	-	-	-	-	-	1	-	-	-	-	-	-	-	1	0.04
*Culex pipiens*	337	14	-	23	-	-	29	3	2	5	-	-	-	413	18.54
*Culex hortensis*	-	-	-	-	-	-	-	-	2	-	-	-	-	2	0.09
*Culex laticinctus*	-	-	-	-	-	-	1	-	-	-	-	-	-	1	0.04
*Culex theileri*	1	-	-	8	-	-	-	-	-	-	2	-	-	11	0.49
*Culex univittatus*	1	-	-	-	-	-	-	-	-	-	-	-	-	1	0.04
*Culex* spp.	-	-	-	-	-	-	-	-	3	-	-	13	-	16	0.72
*Culiseta annulata*	-	-	-	2	-	-	-	-	1	-	-	-	-	3	0.13
*Culiseta longiareolata*	-	-	-	-	13	-	2	1	-	-	-	-	-	16	0.72
*Coquillettidia richiardii*	-	-	-	78	-	-	-	-	-	-	-	-	-	78	3.50
*Uranotaenia unguiculata*	197	-	-	20	-	-	-	-	-	-	-	-	-	217	9.74

FG: Foggia province; TA: Taranto province; MT: Matera province.

**Table 2 pathogens-10-01521-t002:** Natural and artificial larval breeding sites inspected in the study area and positive for mosquitoes.

Species	Tree Hole	Pond	Riverbank	Ditch	Animal Trough	Fountain	Outdoor Bathtub	Tin Jar	Bucket	Poultry Drinking Bowl	Flower Pot Plate
*Anopheles maculipennis* sl		XX	X	X							
*Anopheles plumbeus*	X										
*Aedes albopictus*								XX	X		XX
*Aedes caspius*		X		X			X				
*Aedes detritus^(*)^*		X									
*Aedes geniculatus*	X										
*Aedes vexans^(*)^*				X							
*Culex pipiens*				XX	X		X		XXX	X	
*Culex territans^(*)^*			X								
*Culex theileri*		X									
*Culiseta longiareolata*					X	X	X				
*Uranotenia unguiculata*				X							

Evaluation of larvae/pupae collected during entomological surveys: X < 10; XX = 10–50; XXX > 50. ^(*)^ These mosquito species were found only as larvae.

**Table 3 pathogens-10-01521-t003:** Number of *Anopheles* species collected in 2017, and 2019–2020 by municipality and by capture method.

Year (Month)	Site			Species	Capture Method			Total by Province
	Region	ID	Municipality (Province)		Battery- Powered Aspirator	Human Bait^(*)^	CDC-light Trap	
2017 *(Oct)*	**APULIA**	8	Ginosa (TA)	*An. labranchiae*	95			**165**
		9		*An. labranchiae*	3			
		13		*An. labranchiae*	2		1	
		14		*An. labranchiae*	1		1	
2019 *(Oct)*		8		*An. labranchiae*	5			
		8		*An. maculipennis* sl	1			
		9		*An. labranchiae*			19	
		9		*An. maculipennis* sl			6	
		10		*An. labranchiae*			1	
		11		*An. labranchiae*		1		
		12		NEG				
		13		*An. labranchiae*	25			
		13		*An. maculipennis* sl	4			
2020 *(Jul, Sep)*		1	Lesina (FG)	*An. algeriensis*			475	**606**
		1		*An. labranchiae*	37			
		1		*An. maculipennis* sl	1			
		2	Cagnano Varano (FG)	*An. maculipennis* sl	1			
		2		*An. labranchiae*	1			
		3		NEG				
		4	Ischitella (FG)	NEG				
		5	Manfredonia (FG)	*An. labranchiae*	7			
		5		*An. maculipennis* sl	2			
		6		*An. algeriensis*			19	
		6		*An. labranchiae*	51		8	
		6		*An. maculipennis* sl	4			
		7	Monte Sant’Angelo (FG)	NEG				
2017 *(Oct)*	**BASILICATA**	15	Grottole (MT)	*An. superpictus*			64	**438**
		15		*An. labranchiae*			26	
		15		*An. maculipennis* sl			10	
		16	Ferrandina (MT)	*An. superpictus*			3	
		16		*An. labranchiae*			4	
		16		*An. maculipennis* sl			2	
		17	Bernalda (MT)	*An. superpictus*			2	
2020 *(Jun, Jul)*		15	Grottole (MT)	*An. labranchiae*	7		15	
		15		*An. maculipennis* sl	4			
		16	Ferrandina (MT)	*An. labranchiae*	70		21	
		16		*An. maculipennis* sl	108		101	
		18	Salandra (MT)	*An. plumbeus*		1		
				**Total**	**429**	**2**	**778**	**1209**

FG: Foggia province; TA: Taranto province; MT: Matera province; NEG: negative. (*): Sporadic mosquito collections.

**Table 4 pathogens-10-01521-t004:** Sites selected in the Apulia and Basilicata regions, geographical coordinates, farming activities and year of mosquito collection.

Region	Municipality (Province)	ID Site	Locality	GPS	Animals	Collection Period
**APULIA**	Lesina (FG)	1	Idrovora Lauro	N 41.89921 E 15.56171	Buffaloes	23, 28 July 2020 10–11 September 2020
	Cagnano Varano (FG)	2	Torre del lago	N 41.91045 E 15.75372	sheep and goats	10–11 September 2020
		3	Isola Varano	N 41.91603 E 15.81269	cattle, goats, pigs	
	Ischitella (FG)	4	Torre Varano	N 41.91245 E 15.81312	sheep, goats	10–11 September 2020
	Manfredonia (FG)	5	Siponto	N 41.58341 E 15.87891	buffaloes, horses	8–11 September 2020
		6	Lake Salso	N 41.54439 E 15.86457	donkeys, waterfowl	
	Monte Sant’Angelo (FG)	7	Umbra Forest	N 41.81872 E 15.99323	Horses	8 July 2020
	Ginosa (TA)	8	Pantano - la Tagliata	N 40.42527 E 16.82722	sheep, goats, horse	10–14 October 2017 15–16 October 2019
		9	La Stornara	N 40.45445 E 16.84305	sheep, goats, poultry	10–14 October 2017 15–16 October 2019
		10	Girifalco	N 40.47751 E 16.78694	horses	15 October 2019
		11	Loc. Tessali	N 40.45972 E 16.87722	mixed cattle/sheep	15 October 2019
		12	Contrada Tufarelle	N 40.47751 E 16.86611	sheep, goats	16 October 2019
		13	Pozzo dei Porci	N 40.50139 E 16.80752	sheep, goats, horses	10–14 October 2017 16 October 2019
		14	Malaria cases dwelling	N 40.47055 E 16.78500	-	10v14 October 2017
**BASILICATA**	Grottole (MT)	15	Piano del Monaco	N 40.60430 E 16.33613	cattle, sheep	10–14 October 2017 7–9 July 2020
	Ferrandina (MT)	16	Piano del Buono	N 40.43480 E 16.48579	cattle, sheep, poultry	26–29 October 2017 9–10 June 2020 6–9 July 2020
	Bernalda (MT)	17	Contrada Tarantina	N 40.42038 E 16.82011	cattle, sheep	19–23 October 2017
	Salandra (MT)	18	Salandra Wood	N 40.54722 E 16.33888	wild	8 July 2020

FG: Foggia province; TA: Taranto province; MT: Matera province.

## Data Availability

The data generated and the material used during the current study are available from the corresponding author upon reasonable request.

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
