# Peer review of "Entomological Surveillance in Former Malaria-endemic Areas of Southern Italy"

_pathogens, 2021, doi:10.3390/pathogens10111521_

Round 1

Reviewer 1 Report

This is a rather descriptive manuscript, well written and clear. I have only few minor comments (attached). I am curious to know whether you will use the data to do some statistical analysis with environmental and weather variables. I think that would be a great addition to this MS.

Reviewer 2 Report

Dear Authors,

I have reviewed the article entitled Entomological surveillance in former malaria-endemic areas of Southern Italy. The topic is indeed of relevance and the findings can be of importance. However, methods are not easy to understand/lack a clear scheme and therefore, conclusions seem not supported by the results. Specific issues are listed below:

The main points to clarify are related to section Materials and Methods and specifically to the following aspects:

Mosquitoes were collected:

i) using different methods (e.g. battery powered aspirators, CDC-light traps and “On two fortuitous occasions, human bait collections”). It is not clear if all methods have been used in all sites or not. If not, it should be specified why, more details should be provided and possible implications on the analysis of the results should be discussed.

ii) without a clear schedule. Collections have been performed either in summer and/or in autumn, either in one day or for more consecutive days and in different years. Some mosquito species can be missed in a study area due to these differences (e.g. in site 10 the collection has been performed only in one day and only in October). Also this point should be clarified and discussed.

Moreover, it is not clear if and how specific information on the environmental conditions of each study site (e.g. meteorological conditions, temperature, humidity and precipitation) and on human activity have been collected. It is not clear how data have been (statistically) analyzed to investigate associations between mosquito abundance and environmental conditions and human activities.

Therefore, it is not clear how it was possible to demonstrate if differences in terms of mosquitoes distribution can be ascribed to differences in the sampling procedures or, to different environmental/anthropogenic conditions as stated by the authors (i.e. at lines 285-287 “species composition and population density showed differences related to the variety of biotopes and human activities that characterized the selected sites” and at lines 535-537 “Our findings show the significant receptivity of the investigated areas, where the former malaria vectors labranchiae and An. superpictus can be found at different densities depending on the kind of environment, climatic parameters and anthropic activities”).

Results and Discussion sections should be modified according to the above thoughts.

Other issue to be considered are:

Study area (lines 444-473): a detailed description of the geographical settings is provided but it is not specified how this information has been used for the selection of the study sites. Moreover, these regions seem to be characterized by different ecological features and, therefore, it is not easy to understand the specific environmental conditions of the study sites 1-18.

It is not easy to understand also why some details have been provided and why these information was of importance for the selection of the study site (e.g. at line 473 it is reported that fishing is among the main human activities and it is not clear why this is of importance for the study area selection). All this information is confusing for the readers and I would suggest rewriting this part.  

References should be carefully checked throughout the ms. For instance:

i) Sentence: “The first species, which is still present in Italy in scattered foci, was considered the main vector along the central and southern coasts and in Sicily and Sardinia. The second showed a more limited distribution, mainly along the upper and lower Adriatic coasts and in Sardinia [6,7].”. Refs 6 (White, G.B. Mosq Syst 197810, 3-44.) and 7 (Romi, R.  Eur Mosq 584 19994, 8-10) are dated and therefore, not very appropriate to support that the first species “is still present in Italy”.

ii) Sentence: “In addition, Anopheles superpictus, which is not included in the  maculipennis complex and is currently present in central and southern Italy and in Sicily, has played a vector role in these areas in the past [12].” The ref. 12 (i.e. Romi et al., Emerg. Infect. Dis. 20017, 915-919. doi: 593 10.3201/eid0706.010601) does not demonstrate that A. superpictus is currently present in Italy. Moreover, in the study Romi et al., it was reported that “Residual populations of An. labranchiae and An. superpictus could still be present along the coasts of Abruzzo, Molise (east coast), Campania, and Basilicata (west coast), but no relevant densities have recently been reported.”.

The sentence “Some areas of the country are more susceptible to this risk because they are characterized by both climatic and ecological conditions favourable to the development of malaria vectors.” is too vague and specific references are missing.

Minor revisions:

Although this referee is not a mother tongue, my feeling is that language and style could be improved throughout the text.

For instance:

Table 4: waterfowl and wild animals are reported in the column “Animal farming”. Probably these species were not farmed. Please check.

Line 271: “the significant percentages of An. maculipennis sl and An. algeriensis males”. Please specify the percentage.

Lines 521-523: “PCR reactions were performed using two mosquito legs directly as a template for amplification or using 5 μl of DNA extracted from each individual mosquito by means of the PureLink Genomic DNA Kit or Genejet Genomic DNA Purification Kit (Ther-523 moFischer Scientific). This sentence is not clear. Please rephrase.

Line 206: “There was no livestock, apart from the presence of a few dogs”. Dogs are not livestock.

Round 2

Reviewer 2 Report

Dear Authors,

Thank you for your replies and clarifications. In my opinion, no further revisions are needed.